# ARL3 activation requires the co-GEF BART and effector-mediated turnover

Yasmin ElMaghloob[1], Begoña Sot[2,3†], Michael J McIlwraith[1†], Esther Garcia[1], Tamas Yelland[1], Shehab Ismail[1,4,5]*

[1]CRUK- Beatson Institute, Glasgow, United Kingdom; [2]Fundación IMDEA-Nanociencia, Campus de Cantoblanco, Madrid, Spain; [3]Unidad Asociada de Nanobiotecnología (CNB-CSIC e IMDEA Nanociencia), Campus de Cantoblanco, Madrid, Spain; [4]Institute of Cancer Sciences, University of Glasgow, Glasgow, United Kingdom; [5]Department of Chemistry, KU Leuven, Celestijnenlaan, Heverlee, Belgium

**Abstract** The ADP-ribosylation factor-like 3 (ARL3) is a ciliopathy G-protein which regulates the ciliary trafficking of several lipid-modified proteins. ARL3 is activated by its guanine exchange factor (GEF) ARL13B via an unresolved mechanism. BART is described as an ARL3 effector which has also been implicated in ciliopathies, although the role of its ARL3 interaction is unknown. Here, we show that, at physiological GTP:GDP levels, human ARL3GDP is weakly activated by ARL13B. However, BART interacts with nucleotide-free ARL3 and, in concert with ARL13B, efficiently activates ARL3. In addition, BART binds ARL3GTP and inhibits GTP dissociation, thereby stabilising the active G-protein; the binding of ARL3 effectors then releases BART. Finally, using live cell imaging, we show that BART accesses the primary cilium and colocalises with ARL13B. We propose a model wherein BART functions as a bona fide co-GEF for ARL3 and maintains the active ARL3GTP, until it is recycled by ARL3 effectors.

**\*For correspondence:**
shehab.ismail@glasgow.ac.uk

[†]These authors contributed equally to this work

**Competing interests:** The authors declare that no competing interests exist.

## Introduction

Several ADP-ribosylation factor-like (ARF-like) proteins, such as ARL3 and ARL6, have been found to play important roles in ciliary biogenesis and homeostasis. Defects in these proteins can therefore result in various ciliopathies as ciliary formation and/or composition becomes compromised (*Alkanderi et al., 2018*).

ARL3, and its homologue ARL2, bind the phosphodiesterase six subunit δ (PDEδ) and the uncoordinated paralogues UNC119A and UNC119B. PDEδ and the UNC119 paralogues bind to and are involved in the trafficking of prenylated and myristoylated proteins, respectively. Due to their functional and structural similarities to RHO GDP-dissociation inhibitors (GDIs), they have been named GDI-like solubilising factors (*Chandra et al., 2012*). ARL3 acts as a release factor for these proteins, and is therefore crucial for the trafficking of their lipid-modified cargo (*Ismail, 2017*). In addition, the binder of ARL2 (BART) has been previously identified as an effector of ARL2 and ARL3 (*Sharer and Kahn, 1999*; *Veltel et al., 2008*). Although its functions remain unclear, BART has been implicated in the ciliopathic disorder retinitis pigmentosa, indicating a role in primary cilia (*Davidson et al., 2013*).

As with other GTPases, the activity of ARL proteins is fine-tuned by the action of guanine nucleotide exchange factors (GEFs) and GTPase-activating proteins (GAPs); these accelerate nucleotide exchange and GTP hydrolysis, respectively, thereby modulating G-protein activity. The regulation of ARL3 is critical for ciliary function, as both its reported GEF, ARL13B (*Alkanderi et al., 2018*; *Gotthardt et al., 2015*), and GAP, RP2 (*Schwahn et al., 1998*; *Veltel et al., 2008*), have been implicated in ciliopathies. However, the ARL13B mechanism of action is not fully understood. While it fits

the profile of the ARL3GEF owing to its ciliary localisation, and in vitro GEF activity, the reported ARL3-ARL13B complex structure has not elucidated the interactions between the GEF and the ARL3 nucleotide-binding site (*Gotthardt et al., 2015*), necessary for nucleotide exchange.

Here, we report and describe the human ARL3GEF as a two-component factor comprising ARL13B and BART, both proteins being able to bind nucleotide-free ARL3 and co-localise to the primary cilium. Although the individual proteins failed to efficiently exchange ARL3GDP to ARL3GTP, together, BART and ARL13B were able to completely exchange ARL3 at an accelerated rate of over 500-fold compared to the intrinsic exchange rate. In addition to binding to the nucleotide-free ARL3, BART binds to ARL3GTP and inhibits GTP dissociation. We show that ARL3 effectors displace BART from the BART-ARL3 complex. Thus, we propose a model in which: BART functions as a co-GEF activating ARL3 together with ARL13B; BART then maintains ARL3 in the GTP-bound active form, thereby circumventing the ARL3 preference for GDP, until it is displaced and recycled by ARL3 effectors.

## Results and discussion

### ARL3 is weakly activated by ARL13B under physiological GTP:GDP levels

GEFs generally reduce the binding affinities of small G-proteins to nucleotides, thereby allowing the fast replacement of the bound GDP with GTP, which is more abundant in cells. GEFs are known not to favour the binding to, and hence dissociation of, one nucleotide-bound form of their cognate GTPase over another (*Goody, 2014*). The physiological concentration of GTP is approximately 10 times that of GDP, although this ratio can vary depending on the cell type and status (*Traut, 1994*). For a G-protein with similar binding affinities to GDP and GTP, a 10-fold excess of GTP should result in 90% loading of GTP at equilibrium, assuming no GTP hydrolysis. However, in the case of ARL3, it is reported that the ARL3-binding affinity for GDP is 50 times that of the GTP analogue GppNHp (0.06 nM and 3 nM, respectively [*Veltel et al., 2008*]). This difference in affinity would nullify the 10-fold excess in GTP and would result in a majority of GDP-loaded ARL3 at equilibrium even in the presence of GEFs. ARL13B has been previously identified as a GEF for ARL3, although the mechanism has not been fully elucidated (*Gotthardt et al., 2015*). We hence set out to investigate the GEF activity of ARL13B using GDP and GTP nucleotides.

To investigate the binding of ARL3 to GTP versus GDP in the presence of ARL13B and at physiological GDP:GTP concentrations, we incubated GDP-loaded ARL3 with recombinant ARL13B[18-278], the minimum active ARL13B construct (*Gotthardt et al., 2015*) and nucleotide mixtures of fluorescently-labelled GDP (mantGDP) and 10-fold excess unlabelled GDP, GTP, or GppNHp (*Figure 1A*). The binding of mantGDP to ARL3 was monitored using fluorescence polarisation measurements. As expected, excess unlabelled GDP was able to compete with mantGDP and the binding of mantGDP was almost fully inhibited. However, even in the presence of ARL13B and physiological ratios of unlabelled GTP or GppNHp to GDP, the majority of ARL3 was bound to GDP (*Figure 1A*). We conclude that, in the presence of ARL13B and under physiological relative concentrations of GTP:GDP, ARL3 is weakly activated at equilibrium and the majority of ARL3 (>50%) is loaded with GDP.

The fraction of ARL3 bound to mantGDP is in line with the lower affinity of ARL3 to GTP or GppNHp compared to GDP. The binding of mantGDP in the presence of GTP (53% compared to that in absence of unlabelled nucleotides) is relatively less than that in the presence of GppNHp (83.7% compared to that in absence of unlabelled nucleotides) which reflects a mildly higher affinity of GTP toward ARL3 compared to its analogue GppNHp (*Figure 1A*). However, this difference in affinity between GTP and GppNHp may be due to the hydrolysis of GTP or the contamination of the GTP solution with GDP, which is common in commercial GTP preparations. Therefore, to further investigate the rate and extent of dissociation of GDP, we used excess unlabelled GDP or GppNHp in ARL3mantGDP exchange (*Figure 1B*). Under the conditions used, ARL13B was able to accelerate the dissociation of GDP approximately 8.5-times in the presence of unlabelled GDP and 25-times in the presence of GppNHp compared to the intrinsic rate with GDP alone. This is consistent with the higher activity of ARL13B in the presence of GTP compared to GDP (*Gotthardt et al., 2015*). GppNHp, as expected, was not able to completely displace GDP despite being in 75-fold excess (*Figure 1B*).

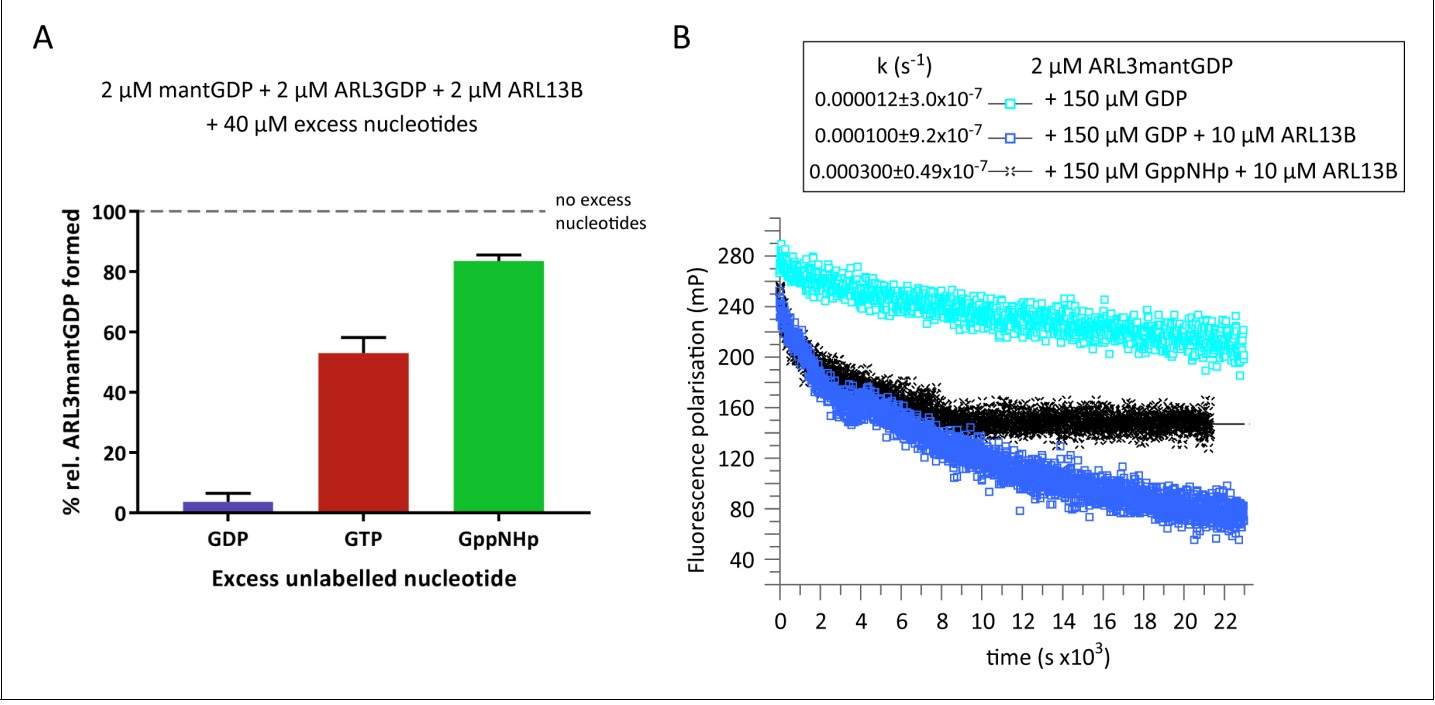

**Figure 1.** ARL3GDP guanine nucleotide exchange is poorly catalysed by ARL13B. (**A**) 2 µM mantGDP was mixed with 2 µM ARL3GDP, 2 µM ARL13B[18-278], and 40 µM unlabelled nucleotide (GDP, GTP, or GppNHp, as indicated). Fluorescence polarisation was measured following the addition of mantGDP to monitor the formation of ARL3mantGDP over time until the stabilisation of the measurements. The experiment was repeated three times and the average maximum polarisation obtained for each nucleotide is shown relative to that obtained with no excess unlabelled nucleotides added to the reaction. (**B**) Fluorescence polarisation measurements were used to monitor the nucleotide exchange of 2 µM ARL3mantGDP in the presence of 150 µM GppNHp (turquoise squares). The exchange was also observed with the addition of 10 µM ARL13B[18-278] in the presence of 150 µM GDP (blue squares) and 150 µM GppNHp (black crosses). Excess unlabelled nucleotides were added to initiate the reaction and the measurements immediately started. The data was fitted using GraFit, and the corresponding exchange rates are listed to the left of the key.

The online version of this article includes the following source data for figure 1:

**Source data 1.** Source data for Figure 1a.
**Source data 2.** Source data for Figure 1b.

We therefore conclude that, under physiological GTP:GDP concentrations and our experimental settings, ARL3 is weakly activated by ARL13B.

## Lipid membranes do not release the ARL3 hasp

ARF subfamily proteins are characterised by the presence of an interswitch toggle and an N- terminal extension, usually an amphipathic helix (*Pasqualato et al., 2002*). The interswitch toggle is located between switch I and switch II and undergoes a nucleotide-dependent movement, which, in the GDP-bound state, allows the retraction of the N-terminal extension on the surface of the protein. In the presence of membranes and upon binding to GTP, the interswitch toggle displaces the retracted N-terminus (*Antonny et al., 1997*; *Goldberg, 1998*). However, the displacement of the amphipathic helix is not possible in the absence of membranes; the helix therefore acts as a hasp, locking ARFs in the GDP-bound form. In line with this, ARFGEFs are not functional in the absence of membranes (*Béraud-Dufour et al., 1999*). Comparing the structures of GTP- and GDP-bound ARL3 shows that it undergoes similar conformational changes as in ARF proteins (*Figure 2A*). We thus wanted to investigate if the ARL3 amphipathic helix acts as a hasp to slow down GDP dissociation, and if this effect can be counteracted by the presence of membranes.

To test the impact of the N-terminal amphipathic on the nucleotide exchange, we used N-terminally truncated ARL3[17-182] (ARL3[ΔN]). In contrast to the full-length protein, ARL3[ΔN] undergoes

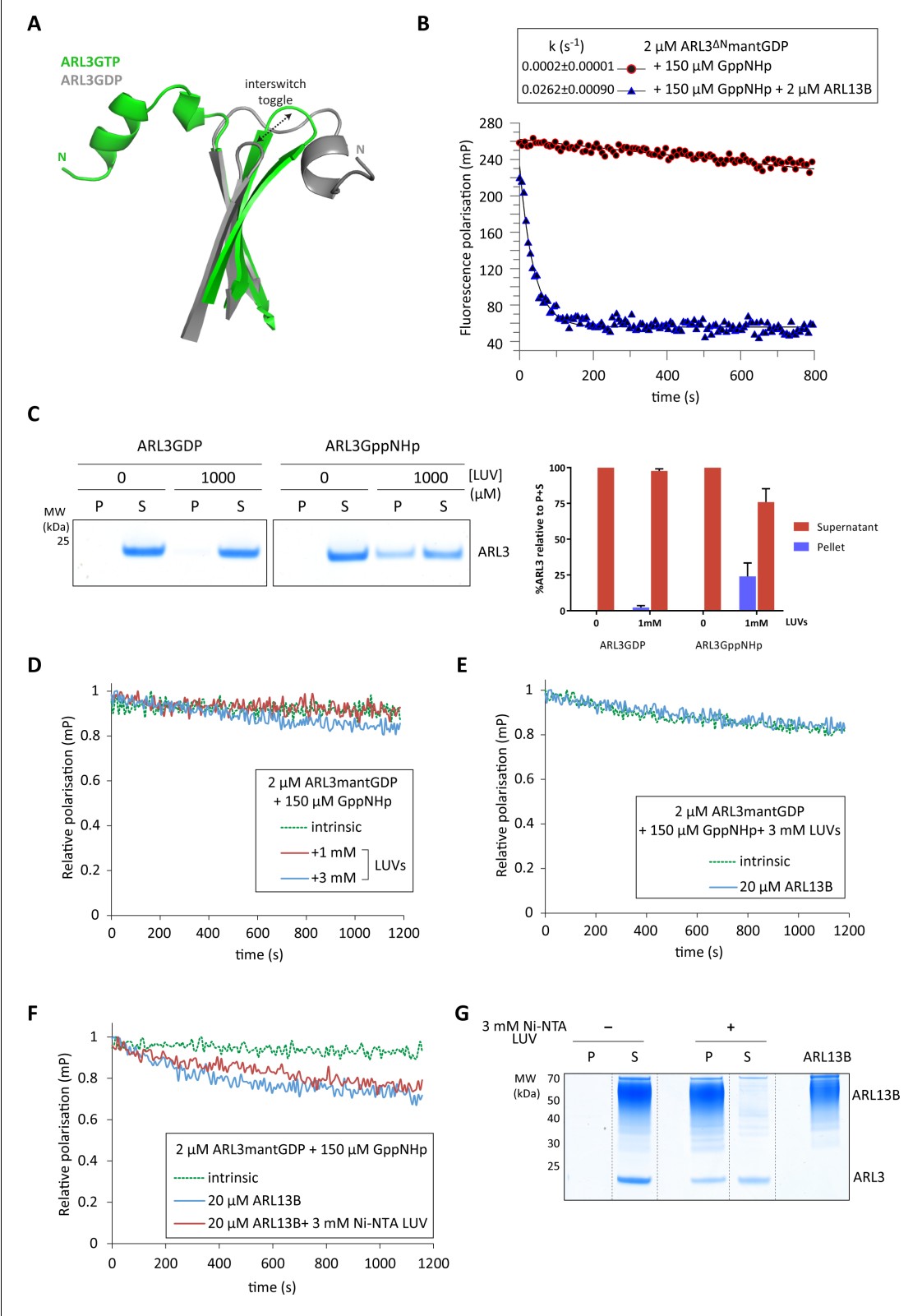

**Figure 2.** Inhibition of ARL3 guanine nucleotide exchange exerted by the N-terminal helix is not released in the presence of liposomes. (**A**) Structure superimposition of the N-terminal helices and the interswitch regions of ARL3GTP (green, PDB: 4ZI2) and ARL3GDP (grey, PDB: 1FZQ) shown in cartoon form. The conformational shift of the interswitch β-hairpin is indicated by the dashed black arrow. (**B**) Nucleotide exchange was monitored for 2 μM N-terminally truncated ARL3mantGDP (ARL3$^{\Delta N}$) mixed with 150 μM GppNHp in the presence (solid blue) and absence (dotted green) of 20 μM

*Figure 2 continued on next page*

*Figure 2 continued*

ARL13B[18-278]. Polarisation was measured after the addition of excess GppNHp. (**C**) Liposome-binding assay in which 2 µM GppNHp- and GDP-loaded ARL3 (top and bottom, respectively) was incubated with increasing concentrations of liposomes (200 nm; DOPC:DOPG:DPPC:DPPG:cholesterol). Following ultracentrifugation, samples of the pellet (**P**) and supernatant (**S**) were run on SDS-PAGE gels. %ARL3 relative to total was quantified for each band. Cosedimentation with 1 mM liposomes was carried out twice with similar results; a representative image of the get is shown here. (**D**) Relative fluorescence polarisation measurements of 2 µM ARL3mantGDP and 150 µM GppNHp in the absence (dotted green) and presence of 1 mM (solid red) and 3 mM (solid blue) liposomes(200 nm; DOPC:DOPG:DPPC:DPPG:cholesterol). Excess GppNHp was added to initiate the reaction and the measurements immediately started. (**E**) The same experiment as (**B**) was repeated with a mixture of 2 µM full-length ARL3, 150 µM GppNHp, and 3 mM liposomes (200 nm; DOPC:DOPG:DPPC:DPPG:cholesterol) in the presence (solid blue) and absence (dotted green) of 20 µM ARL13B[18-278]. (**F**) 2 µM untagged ARL3mantGDP was mixed with 150 µM GppNHp, and fluorescence polarisation was measured in the absence (dotted green) and presence (blue) of 20 µM ARL13B. Nucleotide exchange was also monitored (red) in the presence of 20 µM ARL13B[18-278] and 3 mM of Ni-NTA liposomes (200 nm; DOGS-NTA:DOPC:DOPG:DPPC:DPPG:cholesterol). Excess GppNHp was added to initiate all reactions. (**G**) Liposome-binding assay where 2 µM ARL3GDP was mixed with 150 µM GppNHp and 20 µM 12xHis-ARL13B[18-278] in the presence and absence of 3 mM Ni-NTA liposomes (200 nm; DOGS-NTA:DOPC:DOPG:DPPC:DPPG:cholesterol). Following ultracentrifugation, samples of the pellet (**P**) and supernatant (**S**) were run on SDS-PAGE gels, along with a sample of 12xHis-ARL13B[18-278] for reference. Dashed lines indicate lanes cut for presentation.

The online version of this article includes the following source data for figure 2:

**Source data 1.** Source data for Figure 2b.
**Source data 2.** Source data for Figure 2c.
**Source data 3.** Source data for Figure 2d.
**Source data 4.** Source data for Figure 2e.
**Source data 5.** Source data for Figure 2f.

significant nucleotide exchange upon the addition of ARL13B (*Figure 2B*). This supports the notion of the ARL3 amphipathic helix acting as a hasp, which inhibits ARL13B-mediated nucleotide exchange.

ARL3 differs from ARF proteins in the reported lack of myristoylation of its N-terminus, which implies weak affinity to membranes (*Fansa and Wittinghofer, 2016*). Attempts to carry out in vitro N-myristoyl transferase 1 (NMT1)-catalysed myristoylation of ARL3 (*Stephen et al., 2018*) also did not result in any detected myristoylated N-terminal ARL3 fragments as analysed by mass spectrometry. We thus tested the binding of ARL3 to liposomes previously shown to interact with ARL3 (*Kapoor et al., 2015*; *Lokaj et al., 2015*). Both GDP- and GppNHp-bound ARL3 show weak binding, with ARL3GppNHp being stronger (*Figure 2C*), which is most likely due to the better availability of the N-terminal helix in the GTP-bound conformation. While our experiments show that the ARL3 N-terminal helix cannot drive stable lipid binding, the lipid environment may induce conformational changes in the former, allowing the nucleotide exchange to occur.

We therefore tested whether the presence of liposomes can release the N-terminal helix and so affect the rate of ARL3 nucleotide exchange. The dissociation of mantGDP from ARL3 was monitored in the presence of 1 mM and 3 mM liposomes in the absence and presence of ARL13B[18-278]. Under the experimental conditions used, however, nucleotide exchange was not enhanced (*Figure 2D and E*).

ARL13B itself is palmitoylated in cells (*Cevik et al., 2010*), and its interaction with lipid membranes as well as ARL3 may result in an increased local concentration of the latter on the membrane surface, driving ARL3 N-terminal interaction with lipids and enhancing the dissociation of bound GDP. Palmitoylated ARL13B is difficult to obtain due to its insolubility in aqueous solutions, and the chemical lability of the palmitoyl thioester linkage. Instead, to simulate the ARL13B membrane interaction, we used nickel-chelating liposomes with His-tagged ARL13B[18-278] and monitored the nucleotide exchange of untagged ARL3mantGDP (*Figure 2F*). Dissociation of ARL3-bound mantGDP is similar in the presence and absence of Ni-chelating liposomes, despite the strong liposomal binding of His-tagged ARL13B[18-278] (*Figure 2G*). Overall, under our experimental settings, ARL3 nucleotide exchange in all liposome reactions was much less compared to that of ARL3[ΔN](*Figure 2B*), indicating that the liposomes used failed to release the N-terminal helix.

ARL13B-mediated nucleotide exchange of the full-length ARL3 may therefore require another non-lipid component to the reaction, which is likely to involve the release of the ARL3 N-terminus.

## BART is a co-GEF for ARL13B-mediated ARL3 nucleotide exchange

We reasoned that the release of the ARL3 amphipathic helix may be carried out by one of its interactors. This protein would be involved in the guanine nucleotide exchange reaction and therefore must be able to bind to nucleotide-free ARL3 (*Cherfils and Chardin, 1999*). We thus investigated the ability of ARL3 interactors to bind to the GTPase in various nucleotide states. We performed pull-down assays using BART and GDI-like solubilising factors (represented by UNC119A); UNC119A shows strong specificity toward ARL3GppNHp as expected for an effector. Surprisingly, BART showed weak but consistent binding to nucleotide-free ARL3 (1:10 compared to ARL3GppNHp binding), identifying it as a potential GEF candidate (*Figure 3A*). This is further supported by the reported structure of the ARL3 homologue ARL2-GTP in complex with BART which shows BART-ARL2 N-terminal helix interaction (*Zhang et al., 2009*). Similar interactions were reported for the structure of ARL3 in complex with the related protein BART-like 1 (BARTL1) (*Lokaj et al., 2015*).

We therefore wanted to test the effect of BART on ARL3 activation; the addition of BART to the ARL13B-mediated reaction is twofold: a significant amount of mantGDP is displaced, and the ARL3 nucleotide exchange rate is markedly increased (*Figure 3B*). Titrating in BART to the ARL13B-mediated reaction resulted in a maximum increase of over 500-fold that of the intrinsic rate of GDP dissociation (*Figure 3C*).

To investigate whether this effect occurs due to BART binding to ARL3GTP or to the nucleotide-free form (*Figure 3A*), the GEF assay was repeated using ARL3mantGDP and excess unlabelled GDP. BART was able to accelerate the nucleotide exchange approximately 70-fold (*Figure 3B*), supporting that BART has bona fide GEF activity together with ARL13B, thereby acting as a 'co-GEF'. Furthermore, the addition of ARL3 effectors UNC119A and PDEδ to the ARL13B-mediated reaction failed to enhance the nucleotide exchange of ARL3mantGDP, despite their high affinity to ARL3GTP (*Figure 3D*). Superimposition of the ARL13B-ARL3 structure with that modelled for the ARL3-BART complex reveals no steric clashes and that a ternary complex is structurally feasible (*Figure 3E*).

In addition to the autoinhibition exerted by the N-terminal helix, ARL3GDP, similar to ARF proteins, is further inhibited by the interswitch β-hairpin, which adopts a conformation that blocks GTP binding (*Greasley et al., 1995*). GTP binding requires a conformational shift of the interswitch β-hairpin which then occupies the pocket vacated by the N-terminal helix (*Nawrotek et al., 2016*; *Figure 2A*). These changes must be mediated by the GEF for ARL3 activation to occur. In the reported ARL13B-ARL3 crystal structure, ARL13B does not interact with the ARL3 N-terminal helix or the interswitch region. A comparison of the reported crystal structures of ARL3 in complex with its identified interactors (*Hillig et al., 2000*; *Ismail et al., 2012*; *Hanzal-Bayer, 2002*; *Gotthardt et al., 2015*) shows that BART is the only ARL3 interactor that can interact with both the N-terminal helix and switches including the interswitch region (*Figure 3E*). Furthermore, BART shows stronger binding to the GTP-loaded and nucleotide-free ARL3 compared to the GDP-bound form (*Figure 3A*). We speculated if BART can stabilise the conformational shift of the interswitch region to favour the dissociation of the bound GDP and allowing the binding of GTP.

The presence of BART is able to accelerate and increase the nucleotide exchange of ARL3mantGDP with unlabelled GDP, excluding the necessity of binding to the product, ARL3GTP, to accelerate the dissociation of the bound GDP (*Figure 3B*). To test the involvement of the BART interactions with the switches and the interswitch regions, BART was added to ARL13B-mediated nucleotide exchange of ARL3$^{\Delta N}$mantGDP. Despite the absence of the helix, BART is still able to accelerate the exchange, although a similar amount of ARL3$^{\Delta N}$ is exchanged compared with ARL13B alone (*Figure 3F*). Therefore, BART carries out this novel mode of action as a co-GEF in the ARL13B-mediated reaction by releasing the two layers of autoinhibition.

## ARL3 effectors facilitate BART turnover

The high affinity of BART to ARL3GTP gives rise to the possibility of product inhibition of the former. Indeed, under multiple turnover conditions, where limiting concentrations of BART were added to the ARL13B-mediated reaction, this becomes evident. The presence of excess GppNHp resulted in rapid quenching of the reaction and a small amount of exchange, as the initial production of ARL3GTP leads to the formation of a complex with BART that inhibits the turnover of the reaction. On the other hand, presence of excess GDP did not result in the same inhibition, which is to be

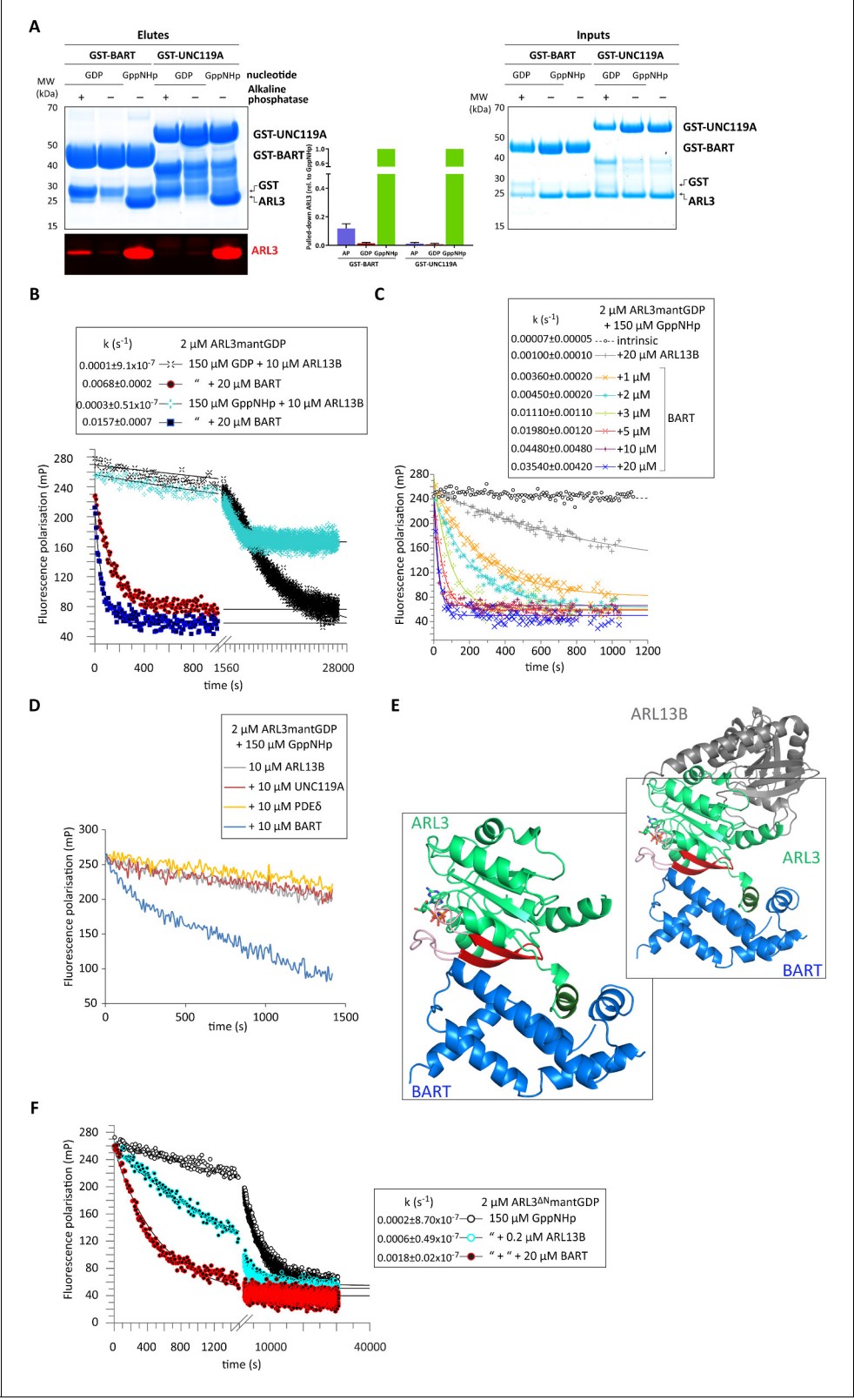

**Figure 3.** BART acts as a co-GEF in the ARL13B-mediated exchange of ARL3, accelerating and enhancing the efficiency of the reaction. (**A**) GST-tagged BART and UNC119A were used to pull down ARL3 (bound to GDP or GppNHp, as indicated) in the presence and absence of alkaline phosphatase (AP). Samples of the elutes and inputs were run on SDS-PAGE gels and visualised using Coomassie blue. Pulled down ARL3 was further confirmed by immunoblotting using anti-ARL3 antibodies (red), and was quantified relative to the respective GST-BART or GST-UNC119A bands (right). The pull-

*Figure 3 continued on next page*

Figure 3 continued

downs were repeated three times; a representative image of the obtained gels and western blots is shown here. (B) 2 µM ARL3mantGDP was mixed with 10 µM ARL13B[18-278] and 150 µM GDP or GppNHp, and the nucleotide exchange was monitored in the presence and absence of 20 µM BART, as indicated. The first 1000s of the measurement following nucleotide addition are shown in greater detail. The data was fitted using GraFit, and the corresponding exchange rates are listed to the left of the key. (C) 2 µM ARL3mantGDP was mixed with 150 µM GppNHp, and subsequent nucleotide exchange was monitored with the addition of 20 µM ARL13B[18-278] and increasing amounts of BART, as indicated. All reactions were initiated by the addition of excess GppNHp.The data was fitted using GraFit, and the corresponding exchange rates are listed (right). (D) 2 µM ARL3mantGDP was mixed with 150 µM GppNHp and 10 µM ARL13B[18-278] and fluorescence polarisation was monitored over time in the absence (grey) and presence of 10 µM UNC119A (red), PDEδ (yellow), or BART (blue). (E) Superimposition of crystal structures of ARL3 (green, with switch regions, interswitch β-hairpin, and the N-terminal helix in light red, red, and dark green, respectively) in complex with ARL13B (grey, PDB: 5DI3), and BART (blue, PDB: 3DOE). As shown in the magnified area denoted by the black squares, only BART can bind both the N-terminal helix and the switch regions. (F) Nucleotide exchange of 2 µM ARL3[ΔN]mantGDP was monitored in the presence of 150 µM GppNHp (black circles), with the addition of 0.2 µM ARL13B[18-278] (turquoise circles), and with the addition of 0.2 µM ARL13B[18-278] and 20 µM BART (red circles). The data was fitted using GraFit, and the corresponding exchange rates are listed.

The online version of this article includes the following source data for figure 3:

**Source data 1.** Source data for Figure 3a.
**Source data 2.** Source data for Figure 3b.
**Source data 3.** Source data for Figure 3c.
**Source data 4.** Source data for Figure 3d.
**Source data 5.** Source data for Figure 3f.

expected as BART would have low affinity to the produced ARL3GDP, and therefore no inhibition occurs (*Figure 4A*).

As other ARL3 effectors such as UNC119A and PDEδ also exhibit a comparably high affinity to ARL3GTP (*Veltel et al., 2008*), we hypothesised that product inhibition of BART can be relieved by the presence of ARL3 effectors which would displace BART from the ARL3GTP-bound complex. The addition of UNC119A in equal concentrations as ARL3 to a GEF assay with limiting amounts of ARL13B and BART resulted in greater ARL3 exchange compared to that obtained in the absence of UNC119A. This enhancement of the exchange was only observed when all three proteins, GEF, co-GEF, and effector, were added, confirming that UNC119A displaces BART from the ARL3-BART complex (*Figure 4B*).

BART interacts with ARL3 via the switches and the N-terminal helix simultaneously (*Figure 3E*). It was shown that both interactions are essential for high-affinity interaction of BART with the ARL3 homologue ARL2 (*Zhang et al., 2009*). This raises the possibility of an avidity effect of the two binding sites. In this case, the inhibition of one binding/interacting site should increase the dissociation rate of BART from ARL3 due to loss of avidity. Since effectors should only compete for the switches, we wanted to test if their binding can accelerate the release, and therefore the recycling, of BART. We investigated the displacement of ARL3GppNHp from immobilised BART using surface plasmon resonance (SPR); as UNC119A exhibited unspecific binding to the used chip, the homologue PDEδ was used instead. Surprisingly, ARL3 was displaced twofold faster with the addition of the ARL3 effector PDEδ supporting the possibility that, ARL3 effectors not only displace BART from ARL3-BART but also accelerate its recycling (*Figure 4C*).

## BART accesses the primary cilium and co-localises with ARL13B

BART has been reported to localise to the connecting cilia of mouse photoreceptors and has been implicated in ciliopathic disease (*Davidson et al., 2013*). However, given the exclusive ciliary localisation of ARL13B (*Caspary et al., 2007*), BART would need to have access to the ciliary compartment to act as a co-GEF in ARL3 nucleotide exchange. Indeed, live cell imaging of ciliated NIH/3T3 and RPE cells shows that BART can access the cilia and co-localise with ARL13B (*Figure 5A*).

BART binds to ARL3GTP and inhibits the dissociation of the otherwise relatively fast dissociating GTP (*Figure 5B*). ARL2 is a homologue of ARL3 that binds to GTP or GDP with relatively weak affinities (*Veltel et al., 2008*). Due to the weak binding affinities and the fast dissociation of the bound nucleotide, it has been proposed that ARL2 does not need a GEF and undergoes rapid nucleotide exchange on its own (*Fansa and Wittinghofer, 2016*). However, BART is able to bind to ARL2GTP

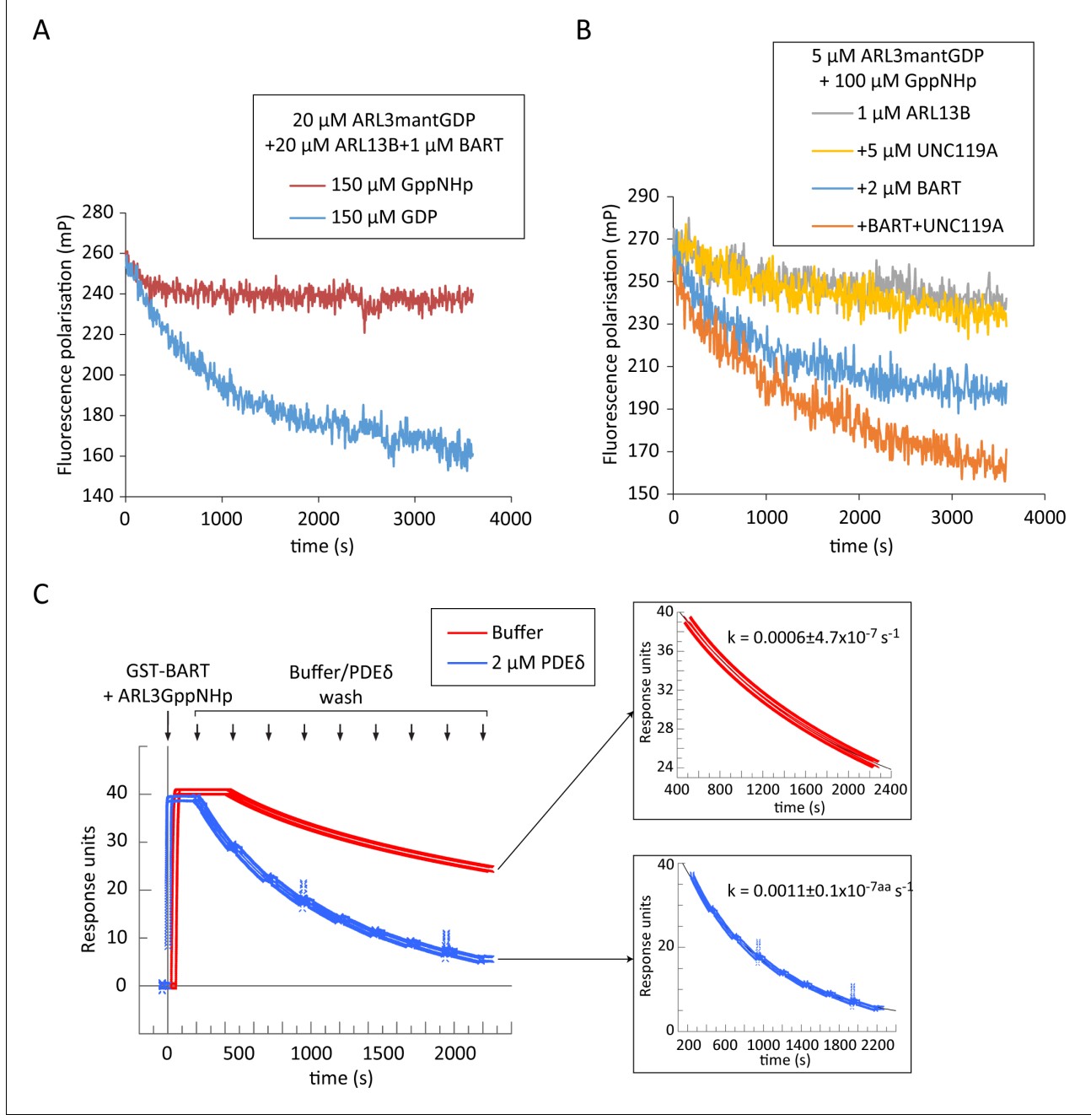

**Figure 4.** BART product inhibition is relieved by ARL3 effectors which mediate its turnover. (**A**) Nucleotide exchange of ARL3 was assayed in a multiple turnover experiment with a limiting addition of BART. Fluorescence polarisation was measured following nucleotide addition for a mixture of 20 μM ARL3mantGDP, 20 μM ARL13B, and 1 μM BART in the presence of either 150 μM GppNHp (red) or 150 μM GDP (blue). (**B**) Fluorescence polarisation was measured for a mixture of 5 μM ARL3mantGDP, 100 μM GppNHp, and 1 μM ARL13B[18-278] (grey). The measurements were repeated in the presence of 5 μM UNC119A (yellow), 2 μM BART (blue), or a mixture of UNC119A and BART at the aforementioned concentrations (orange). All reactions were initiated by the addition of GppNHp. (**C**) SPR sensorgrams of ARL3GppNHp (640 nM) bound to immobilised GST-BART undergoing nine washing steps (black arrows) with either running buffer (red), or with 2 μM PDEδ (blue). The data was fitted to a single exponential equation using GraFit, and the fitted curves are shown on the right, along with the corresponding rate constants.

The online version of this article includes the following source data for figure 4:

**Source data 1.** Source data for Figure 4a.
**Source data 2.** Source data for Figure 4b.
**Source data 3.** Source data for Figure 4c.

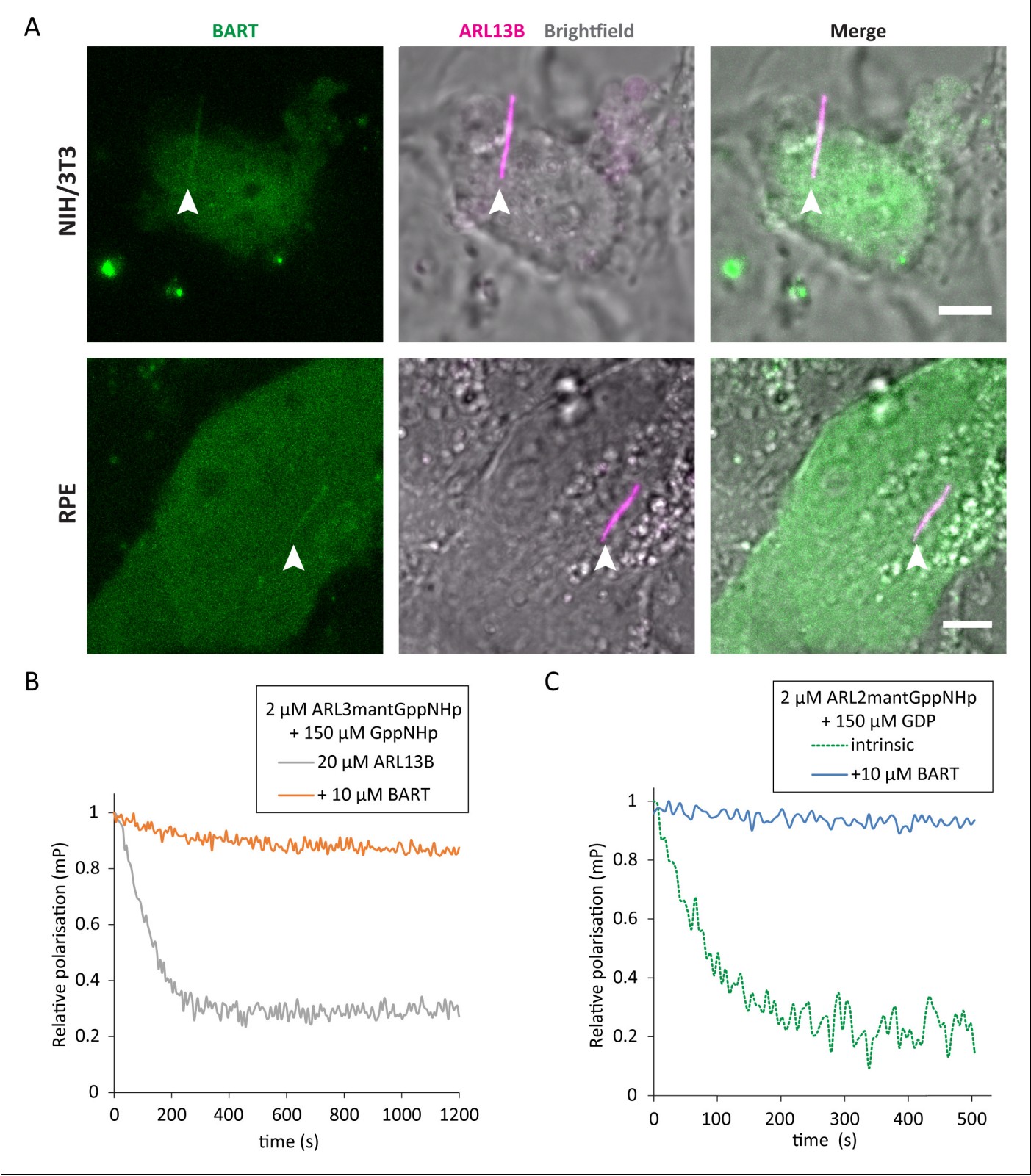

**Figure 5.** BART localises to the primary cilium and maintains active GTP-bound forms of ARL3 and ARL2. (**A**) Representative images of live NIH/3T3 (top) and RPE (bottom) cells transfected with pEGFP-C1(BART) and pmKate2-N(ARL13B) and serum starved for 24 hr. The images are maximum projections of three optical slices. Transfections and subsequent imaging was carried out at least two times. Scale bars, 5 μm; white arrows indicate cilia. (**B**) Fluorescence polarisation measurements of 2 μM ARL3mantGppNHp, 150 μM GppNHp, and 20 μM ARL13B[18-278] in the absence (grey) and

*Figure 5 continued on next page*

Figure 5 continued

presence (orange) of 10 µM BART. GppNHp was added last to both reactions before starting measurements. (**C**) 2 µM ARL2mantGppNHp was mixed with 150 µM GDP, and the resultant nucleotide exchange following GDP addition was monitored in the absence (dotted green) and presence (blue) of 10 µM BART.

The online version of this article includes the following source data for figure 5:

**Source data 1.** Source data for Figure 5b.
**Source data 2.** Source data for Figure 5c.

and inhibit the dissociation of GTP (*Figure 5C*). We therefore propose that BART can maintain an active pool of ARL3GTP and ARL2GTP in the cell.

## Discussion

Generally, GEFs generally accelerate the dissociation of nucleotides bound to their cognate G-proteins. GTP and GDP then compete for the nucleotide-free G-protein. In cells, where GTP is more abundant than GDP (10-fold), and in the presence of GEFs the product will be predominantly GTP-loaded and GEFs are adequate to activate their cognate G-proteins (*Traut, 1994*; *Goody, 2014*). Nevertheless, this scenario is only valid assuming that the G-protein of interest has equal or similar binding affinities to GTP and GDP. Here we show that, for ARL3 where the binding affinities for GTP and GDP differ significantly (*Veltel et al., 2008*), ARL13B is not able to efficiently activate it. Our study underscores the importance of taking into consideration the binding affinities of small G-proteins to GTP and GDP while investigating their mechanism of activation. We argue that using non-equivalent nucleotides in their physiological relative GTP:GDP concentrations in standard GEF assays is crucial to fully characterise and understand the mechanisms of activation.

GTP binding of ARF and ARL proteins requires the release of the autoinhibition exerted by the N-terminal helix and the conformation of the interswitch region. For ARF proteins, this hurdle is circumvented by the binding of the myristoylated helix to membranes (*Nawrotek et al., 2016*). As previously discussed, differences in ARL lipidation mean that the activation of ARL proteins requires a different solution.

Our model provides a potential answer to the question of how ARL3 nucleotide exchange can take place in the absence of a myristoyl membrane anchor and weak membrane binding. We propose that BART acts in place of lipid membranes in this process, masking the N-terminal amphipathic helix from unfavourable exposure to the aqueous surroundings, both during and after nucleotide exchange, as well as stabilising the interswitch region in the GTP conformation. The formation of a high-affinity BART-ARL3GTP complex following the reaction maintains this unfavourable conformation in solution until the binding of ARL3 effectors (*Figure 6*).

GDP-bound ARL3 (light green box) adopts a conformation in which the N-terminal amphipathic helix (red solid line) is masked from the surroundings by a hydrophobic pocket on the surface of the protein. In addition, the interswitch β-hairpin conformation (light green sold line) clashes with effector binding. In the primary cilium, the binding of BART and ARL13B (anchored to the membrane via palmitoyl moieties, blue solid lines) facilitates the formation of nucleotide-free ARL3 (grey box); the conformational changes in the N-terminal helix and interswitch regions are stabilised by their interactions with BART. The binding of GTP to ARL3 (dark green box) is favoured in the BART-ARL3 complex due to the masking of the exposed N-terminal helix and the stabilisation of the interswitch β-hairpin GTP conformation (dark green solid lines). This high-affinity BART-ARL3 complex maintains the active ARL3 in the aqueous environment, until the displacement of ARL3GTP by effectors such as UNC119 or PDEδ (brown box), releasing BART to further catalyse the ARL13B-mediated nucleotide exchange.

BART and ARL13B are not known to be part of a stable multimeric complex, unlike, for example, TRAPP complexes which acts as RAB GEFs in yeast (*Sacher et al., 1998*; *Sacher et al., 2001*). We therefore propose the term co-GEF to describe the novel mode of action of BART in ARL13B-mediated activation of ARL3.

The ARL3 N-terminal helix was also shown to play a significant role in ARL3 function acting as a 'pocket opener' for UNC119A (*Ismail et al., 2012*); a simultaneous interaction with membranes and

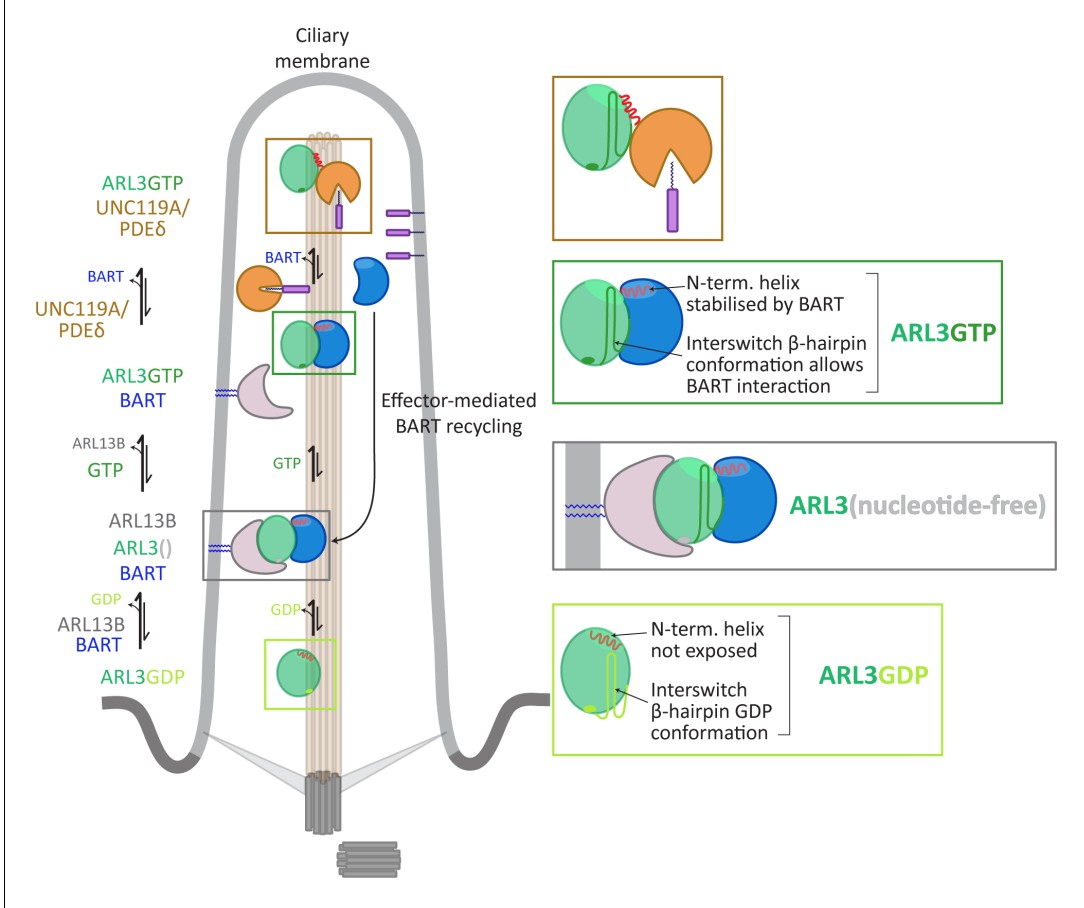

**Figure 6.** Model for co-GEF functionality in ARL13B-mediated nucleotide exchange of ARL3.

UNC119A would not be possible. Indeed, in the structure of ARL3-UNC119A, the N-terminal helix is not exposed to the solution and instead is found retracted on the surface of the protein. All these observations call into question the interaction of ARL3-GTP with cellular membranes. We expect that if ARL3-GTP interacts with membranes the interactions would be very transient and difficult to detect using conventional imaging approaches. Alternatively, ARL3 may form stronger interactions with specific membrane domains in the cell.

The finding of the co-GEF BART may provide clues to other unanswered questions of ARL GTPase activation. One such case is IFT27, which was proposed to be an ARL6GEF based on cell studies and binding to nucleotide-free ARL6. Nevertheless, IFT27 did not show GEF activity in vitro (*Liew et al., 2014*), which raises the possibility of the requirement of a co-GEF as in the case of ARL3.

## Materials and methods

**Key resources table**

| Reagent type (species) or resource | Designation | Source or reference | Identifiers | Additional information |
|---|---|---|---|---|
| Strain, strain background (*Escherichia coli*) | BL21(DE3) CodonPlus (DE3)-RIL | Agilent Technologies | 230245 | Electrocompetent cells |

*Continued on next page*

*Continued*

| Reagent type (species) or resource | Designation | Source or reference | Identifiers | Additional information |
|---|---|---|---|---|
| Cell line (*Homo sapiens*) | hTERT-RPE (immortalised retinal pigmented epithelial cells; female) | ATCC | CRL-4000 | |
| Cell line (*Mus musculus*) | NIH/3T3 (embryonic fibroblasts) | ATCC | CRL-1658 | |
| Antibody | anti-ARL3 (rabbit, polyclonal) | Proteintech | 10961–1-AP | WB (3:5000) |
| Antibody | IRDye 680RD anti-rabbit IgG (donkey) | Li-Cor | 926–68073 | WB(1:5000) |
| Recombinant DNA reagent | pET20b(12-His-ARL13B$^{18-278}$) | This paper | | N-term. His-tagged ARL13B$^{18-278}$ for bacterial expression |
| Recombinant DNA reagent | pET20b(ARL3$^{17-182}$) | This paper | | C-term. His-tagged ARL3$^{\Delta N}$ for bacterial expression |
| Recombinant DNA reagent | pET20b(ARL3) | This paper | | C-term. His-tagged ARL3 for bacterial expression |
| Recombinant DNA reagent | pET20b(ARL3-thrombin-His-tag) | This paper | | Untagged ARL3 for bacterial expression |
| Recombinant DNA reagent | pET20b(BART) | This paper | | C-term. His-tagged BART for bacterial expression; *E. coli* codon-optimised sequence |
| Recombinant DNA reagent | pGEX-4T-1 (BART) | This paper | | N-term. GST-tagged BART for bacterial expression; *E. coli* codon-optimised |
| Recombinant DNA reagent | pGEX-4T-1 (UNC119A) | This paper | | N-term. GST-tagged UNC119A for bacterial expression |
| Recombinant DNA reagent | pET20b (UNC119A) | This paper | | C-term. His-tagged UNC119A for bacterial expression |
| Recombinant DNA reagent | pET20b(PDEδ) | This paper | | C-term. His-tagged PDEδ for bacterial expression |
| Recombinant DNA reagent | pGFP-C1(BART) | This paper | | N-term. GFP-tagged full-length BART for mammalian expression |
| Recombinant DNA reagent | pmKate2-N (ARL13B) | This paper | | C-term. mKate2-tagged full-length ARL13B for mammalian expression |
| Peptide, recombinant protein | Thrombin | Sigma | T4648 | |
| Commercial assay or kit | GST capture kit | Cytiva | BR100223 | |
| Chemical compound, drug | mantGDP | BioLog Life Science Institute | M041-05 | |
| Chemical compound, drug | mantGppNHp | Jena Bioscience | NU-207 | |
| Chemical compound, drug | GppNHp | Jena Bioscience | NU-401 | |

*Continued on next page*

*Continued*

| Reagent type (species) or resource | Designation | Source or reference | Identifiers | Additional information |
|---|---|---|---|---|
| Chemical compound, drug | GTP | Sigma | G8877 | |
| Chemical compound, drug | GDP | Sigma | G7127 | |
| Software, algorithm | GraFit | Erathicus software | | |

Sources of all other chemicals/reagents used are listed in the relevant sections below.

## Cloning, expression, and purification of proteins

Full-length *ARL3* and N-terminally truncated *ARL3*$^{17-182}$ (*ARL3*$^{\Delta N}$) were cloned into pET20b (Novagen) with C-terminal 6-His tags; site-directed mutagenesis was used to insert a thrombin site upstream of the His-tag in the full-length ARL3 construct. *ARL13B*$^{18-278}$ was cloned into pET20b with an N-terminal 12-His tag. Full-length *BART* was codon-optimised (Integrated DNA Technologies) and cloned into pET20b with a C-terminal 12-His tag, as well as pGEX-4T-1 (GE Healthcare) with an N-terminal GST-tag. Full-length UNC119A and PDEδ were also cloned into pET20; UNC119A was also cloned into pGEX-4T-1. pET28(NMT1) was a gift from Jim Brannigan; NMT1 was purified as previously described (*Padovani et al., 2013*).

All proteins were expressed in BL21-CodonPlus (DE3)-RIL competent cells (Agilent Technologies) at 16°C following induction with 250 μM Isopropyl β-D-1-thiogalactopyranoside (IPTG).

ARL3 proteins were purified as previously described (*Kühnel et al., 2006*). ARL13B$^{18-278}$ was purified using a similar method with the addition of 20% glycerol to all buffers. ARL3 digestion to remove the His-tag was carried out overnight at 4°C in elution buffer in the presence of 11 U/mL thrombin (Sigma) and 2 mM CaCl$_2$.

Cells expressing BART, UNC119A or PDEδ were lysed in a buffer comprising 25 mM Tris (pH 7.5), 300 mM NaCl, and 2 mM β-mercaptoethanol. His-tagged and GST-tagged proteins were purified using HisTrap HP and GSTrap HP columns (GE Healthcare), respectively. Proteins were then further purified using size exclusion chromatography and stored in a buffer containing 25 mM Tris (pH 7.5), 150 mM NaCl, and 2 mM DTT. For purification of BART, 5% glycerol was added to all buffers.

## Antibodies and western blots

Anti-ARL3 polyclonal antibody (Proteintech, 10961–1-AP) was used along with IRDye 680RD Donkey anti-Rabbit IgG secondary antibody (Li-Cor, 926–68073). Western blots were scanned using a Li-Cor Odyssey CLX imaging system.

## Nucleotide exchange

A total of 400 μM ARL3 was mixed with 2 mM GppNHp (Jena Bioscience Gmbh) and 69 U alkaline phosphatase (Roche), or with 2 mM GDP and 50 mM EDTA. For loading with mant-labelled GDP (BioLog Life Science) or GppNHp (Jena Bioscience Gmbh), ARL3 was mixed with the appropriate nucleotide at a ratio of 1:2 in the presence of 50 mM EDTA. All reactions were incubated overnight at 19°C. Reactions containing EDTA were stopped by the addition of 100 mM MgCl$_2$. Excess unbound unlabelled or labelled nucleotides were removed using a Superdex 200 Increase column or a PD-10 de-salting column (GE Healthcare), respectively.

## Myristoylation

Myristoylation of ARL3 was carried out in vitro using recombinant N-myristoyl transferase 1 (NMT1) and myristoyl-Coenzyme A (Sigma, M4414) as previously described (*Padovani et al., 2013*).

## Pull-down assays

Pull-down assays were typically done using 30 µg of GST-tagged BART and 30 µg ARL3 in a buffer containing 25 mM Tris (pH 7.5), 200 mM NaCl, and 2 mM DTT. Reactions were incubated for 30 min before addition to glutathione sepharose 4 FF beads (GE Healthcare, 17-5132-01) and incubated for a further 30 min. Beads were washed five times in buffer and proteins were then eluted by incubating the beads in buffer containing 20 mM glutathione for 10 min (Fisher Scientific, 11483074). All incubation steps were done at room temperature. The eluted proteins were run on SDS-PAGE and visualised using Coomassie blue.

## Fluorescence polarisation-based GEF assays

GEF reactions were performed in 25 mM Tris (pH 7.5), 150 mM NaCl, 5 mM $MgCl_2$ and 2 mM DTT at room temperature. Fluorescence polarisation was measured using a Spark multimode microplate reader (Tecan) with excitation and emission wavelengths set at 366 nm and 450 nm, respectively. Typically, 2 µM ARL3mantGDP was mixed with 150 µM GppNHp and 20 µM ARL13B in the presence and absence of 20 µM BART (or as indicated in figure legends) and subsequent changes in fluorescence polarisation were monitored over time. Data analysis was carried out using GraFit (Erathicus Software).

## Preparation of large unilamellar vesicles (LUVs)

1,2-Dioleoyl-sn-glycero-3-phosphocholine (DOPC), 1,2-dipalmitoyl-sn-glycero-3-phosphocholine (DPPC), 1,2-dioleoyl-sn-glycero-3-[phospho-rac-(3-lysyl(1-glycerol))] (DOPG), 1,2-dipalmitoyl-sn-glycero-3-phospho-(1'-rac-glycerol) (DPPG) and 1,2-dioleoyl-sn-glycero-3-{[N-5-amino-1- (carboxylpentyl) iminodiacetic acid]succinyl} (nickel salt) (DOGS-NTA-$Ni^{2+}$) were purchased from Avanti Polar Lipids. Cholesterol was purchased from Sigma Aldrich.

10:5:45:5:20 molar ratio of DOPC:DOPG:DPPC:DPPG:Cholesterol or 10:10:5:45:5:20 molar ratio of DOGS-NTA:DOPC:DOPG:DPPC:DPPG:Cholesterol lipid mixtures in chloroform were dried under $N_2$ stream and rehydrated in a buffer containing 5 mM $MgCl_2$, 150 mM NaCl, and 25 mM Tris·HCl (pH 7.5). The suspension was then subjected to ten cycles of freezing and thawing (at 65˚C) and passed 20 times through a 200 nm filter (Whatman), using the Avanti Mini-Extruder (Avanti Polar Lipids), at 65˚C.

## Cosedimentation assays

2 µM GppNHp- and GDP-loaded ARL3 was incubated with increasing concentrations of DOPC:DOPG:DPPC:DPPG:cholesterol liposomes for 30 min at room temperature in a buffer containing 20 mM Tris (pH 7.5), 150 mM NaCl, 5 mM $MgCl_2$ and 2 mM DTT. Samples were then spun at $125 \times 10^3 \times g$ at 10˚C for 1.5 hr in a TLA-100 rotor (Beckman Coulter). The supernatants were then transferred to new tubes and the pellets resuspended in the aforementioned buffer to an equivalent volume as the supernatants. Equal sample volumes of all pellets and supernatants were analysed by SDS-PAGE.

## Surface plasmon resonance (SPR)

SPR experiments were carried out on a Biacore 8K system (Cytiva) using a CM5 sensor chip (Cytiva), along with a GST capture kit (Cytiva) following manufacturer instructions. The system was maintained at 25˚C, with a flow rate of 30 µL/min. GST or GST-BART was immobilised on the chip, giving a response of about 100 units. 640 nM ARL3GppNHp was then injected for 120 s, followed by nine injections of buffer (20 mM Tris (pH 7.5), 150 mM NaCl, 5 mM $MgCl_2$, 0.02% triton X-100, and 2 mM DTT) or 2 µM PDEδ. The sensor chip was regenerated between cycles with two 30 s injections of 100 mM glycine (pH 2.0) followed by a wash step. The data was analysed using GraFit (Erathicus Software), in which exponential decay was fitted to a single exponential equation with an offset.

## Cell line culture, transfection, and live-cell imaging

NIH/3T3 and RPE cells (both lines newly purchased from ATCC and certified as mycoplasma-free) were grown in DMEM and DMEM:F12 media (Thermo Fisher Scientific), respectively, at 37˚C and 5% $CO_2$. Culture media was supplemented with 10% fetal bovine serum, 1% penicillin-streptomycin, and 2 mM L-glutamine (ThermoFisher Scientific). A total of $2 \times 10^5$ cells were seeded in six-well plates

and grown to 75% confluency prior to transfection with lipofectamine 2000 reagent (ThermoFisher Scientific). Cells were then incubated for another 24 hr and transferred to eight well ibiTreated μ-Slides (Ibidi) with $2 \times 10^5$ cells/chamber. Following adhesion, cells were washed with PBS and serum starved in 0.5% serum-media until imaging 24 hr later.

Fluorescence microscopy was performed using an LSM 880 (Zeiss) fitted with a Plan-Apochromat 63x/1.4NA oil DIC M27, with the samples being placed in a temperature-controlled chamber set at 37°C and 5% $CO_2$. Three 1 μm optical sections with a 0.5 μm interval were taken of the primary cilium, using ARL13B-mKate2 as a marker, and maximally projected.

## Acknowledgements

We thank Alfred Wittinghofer, Wim Versees and Raphael Gasper for the excellent scientific discussions. This work was supported by Cancer Research UK (CRUK core funding award A19257).

## Additional information

### Funding

| Funder | Grant reference number | Author |
|--------|------------------------|--------|
| Cancer Research UK | A17196 | Shehab Ismail |

The funders had no role in study design, data collection and interpretation, or the decision to submit the work for publication.

### Author contributions

Yasmin ElMaghloob, Conceptualization, Data curation, Formal analysis, Investigation, Writing - original draft, Writing - review and editing; Begoña Sot, Conceptualization, Methodology, Writing - review and editing; Michael J McIlwraith, Resources, Data curation, Writing - review and editing; Esther Garcia, Conceptualization, Writing - review and editing; Tamas Yelland, Methodology, Writing - review and editing; Shehab Ismail, Conceptualization, Supervision, Funding acquisition, Validation, Investigation, Writing - original draft

### Author ORCIDs

Yasmin ElMaghloob ![ORCID] https://orcid.org/0000-0002-5613-0678
Shehab Ismail ![ORCID] https://orcid.org/0000-0002-4150-1077

### Decision letter and Author response

Decision letter https://doi.org/10.7554/eLife.64624.sa1
Author response https://doi.org/10.7554/eLife.64624.sa2

## Additional files

### Supplementary files

• Transparent reporting form

### Data availability

All data generated or analysed during this study are included in the manuscript and supporting files. Source data files have been provided for: Figures 1A and 1B, Figures 2B, 2C, 2D, 2E, and 2F, Figures 3A,3B, 3C, 3D, and 3F, Figures 4A, 4B, and 4C and Figures 5B, and 5C.

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
