## [Decision Letter]

**Acceptance summary:**

Arl3 is an important G-protein functioning in ciliary targeting of lipidated cargoes. Arl3 exists in an inactive-GDP loaded form and an active GTP-loaded form. It was previously shown that Arl13B, a ciliary membrane associated protein, is the GEF for Arl3 required for its activation. The authors provide important insights into Arl3 activation by showing that the protein BART acts as a co-GEF that together with Arl13B mediates the efficient activation of Arl3. This represents a significant advance in our understanding of the cycle of Arl3-mediated ciliary delivery of lipid-modified cargo proteins.

**Decision letter after peer review:**

Thank you for submitting your article "ARL3 activation requires the co-GEF BART and effector-mediated turnover" for consideration by *eLife*. Your article has been reviewed by three peer reviewers, and the evaluation has been overseen by Suzanne Pfeffer as the Senior and Reviewing Editor. The reviewers (in alphabetical order) were: Christopher Fromme, Aymelt Itzen and Esben Lorentzen.

The reviewers have discussed the reviews with one another and the Reviewing Editor has drafted this decision to help you prepare a revised submission.

Summary:

Small GTPases are general regulators of intracellular signaling. The small GTPase Arl3 is thought to regulate signaling processes in cilia. Like other small GTPases, Arl3 is inactive when bound to GDP and active when binding to GTP. Guanine nucleotide exchange factors (GEFs) are required to exchange the tightly bound GDP for GTP. In this work, the authors aim to characterize the role of the protein BART in the nucleotide exchange reaction. Based on fluorescence polarization experiments that monitor the release of fluorescently labeled GDP from a complex with Arl3, the activity of BART is characterized. The authors find that full GEF-activity is achieved only in the presence of the previously reported Arl3-GEF Arl13B. The authors conclude that BART acts as a co-GEF operating in concert with Arl13B.

The reviewers are experts and suggested a number of experiments to improve the quality of the story as follows.

Essential revisions:

As described in detail below, the authors should provide more quantitation: they should determine reaction rates and also provide statistics for replicates. Important will be to use a single nucleotide for the measurements as chemically distinct nucleotides will have different affinities (see below).

They should improve Figure 1 by providing more experiments, including GDP/GDP exchange and also the EDTA exchange reactions.

They should improve Figure 3 by also testing reduced concentration of Arl13.

They should improve the Discussion.

Here are more Specific comments to guide your next steps:

1) The authors often make (semi)quantitative statements based on their fluorescence curves and nucleotide release assays, yet they do not provide definite numbers that could be obtained from fitting the results (e.g. Figure 1A and B). This should be performed in order to make the effects of the experiments comparable to each other. In fact, I am not sure that there is an actual difference in nucleotide release rates in Figure 1A and B: The degree of exchange is certainly different for mantGDP and mantGppNHp and this can be explained by the different affinities since GDP binds ~50x more tightly to Arl3 than GppNHp. However, looking at the time it takes for the reactions to reach a plateau, they appear to do so at similar rates. A quantification and the use of equivalent nucleotides would be important to reliably estimate the effects.

2) The Discussion should include a section describing the mechanism of action of other small GTPase GEFs with respect to the dependence on co-GEFs. In particular, the authors should discriminate between the definition of GEFs composed of several subunits and co-GEFs. What is the definition of a co-GEF? Is BART the first example of a co-GEF?

3) If they authors are trying to reproduce physiological GTP/GDP ratios, they should make sure that also unlabeled GDP is available in the buffer at its approximate physiological concentration.

4) I am concerned about the use of non-equivalent nucleotides in the nucleotide-release assays. E.g., in Figure 1A the authors displace mant-GDP with GppNHp. Given that the affinities for GDP and GTP/GppNHp differ by a factor of ~50, equivalent nucleotides should be used for the displacement (e.g. displacing mant-GDP with GDP). Otherwise, the degree of exchange may be dependent on the relative affinities of GDP and GTP for the GTPase. Furthermore, full exchange/displacement should be indicated for every experiment (e.g. by adding EDTA and thus accelerating exchange).

5) Introduction: Describe lipidation of Arl. Explain what the physiological function of GDI is.

6) Introduction: "In addition, the binder of ARL2 (BART) has been previously identified as an effector of ARL2 and ARL3": Please provide reference for this claim.

7) Results: "The addition of recombinant ARL13B18-278…": Please explain why this construct has been used for the experiments. It is mentioned here for the first time. Provide reference to this construct if applicable.

8) Results "GEFs are known not to favour the binding to, and hence dissociation of, one nucleotide-bound form of their cognate GTPase over another." Please provide reference for this claim.

9) Results: "Palmitoylated ARL13B is difficult to obtain due to its instability": Please explain in which sense Arl13B is unstable.

10) Results: "The addition of recombinant ARL13B18-278 to GDP-loaded ARL3 in the presence of up to 250-fold…" Please indicate properly when mant-nucleotides have been used for monitoring exchange reactions. In this instance, the use of mant-nucleotides is not indicated explicitly.

11) Figure 2A: The quality of the depiction should be improved. In particular, the mix in ribbon (zig-zag loops) and cartoon representation is difficult to grasp. Also, light green and green depictions do not sufficiently discriminate the structures from each other.

12) Figure 2C: Statistical errors and test of statistical significance should be included.

13) Figure 3A: The input should be shown in order to evaluate potential changes of the proteins in the course of the experiment. It appears as if Arl3 is not pure, is this true? Furthermore, there is an additional band in the sample GDP+Alk. Phosphatase with GST-BART that disappears in the other lanes. Why is this? I also think that bar diagrams from such experiments should contain error bars. From the Coomassie-stained gel, there appears to be almost no difference in the intensities of Arl3 (if Arl3 is the band with lowest Mw), yet there are large differences in the WB. How can the authors explain this observation?

14) Figure 3B: There appears to be a discrepancy with the figure legend since the figure mentions GppNHp but the text refers to GDP.

15) Figure 3C-H: Why have the experiments been conducted at differing BART and Arl13B concentrations? In my opinion, the concentrations should be kept comparable as much as possible (except for titrations, of course). Otherwise, the observed differences cannot be compared directly.

16) Figure 5 appears to contain data. As such, it should not be part of the Discussion but part of the Results section.

17) Discussion: "Due to the weak binding affinities and the fast dissociation of the bound nucleotide, it has been proposed that ARL2 does not need a GEF and undergoes rapid nucleotide exchange on its own.": Please provide reference for this claim.

18) There should be no "data not shown" in the manuscript.

Reviewer #2:

It would be nice to see some experimental evidence for the Arl3-Arl13B-BART complex modeled in Figure 3G. Alternatively, the authors could design point mutations to disrupt the ARL3-BART interaction and show an effect using their GEF assay.

Reviewer #3:

1) A full understanding of the "physiological" nucleotide dependence of the exchange experiments shown in Figure 1 would include two additional comparisons:

A) The experiment in Figure 1A should be repeated to show what happens if unlabeled GDP is added at time 0, as this addresses the question of whether GDP can efficiently replace mantGDP. Judging by the result shown in Figure 3E, GDP is not able to efficiently replace mantGDP. This result should be discussed in the first part of the Results section to more clearly explain that the kinetic observations of nucleotide preference are due to an inability of Arl13 to stimulate release GDP. Otherwise, the impression given from the results in Figure 1 is that GDP might be able to displace either mantGDP or mantGMP-PNP in the presence of Arl13.

B) For some GTPases, GMP-PNP does not bind nearly as well as GTP. The unlabeled GMP-PNP experiments in Figure 1 should therefore be repeated using unlabeled GTP.

2) In Figure 3, the GEF experiments are described as "multiple turnover" because BART is 1/20th the concentration of Arl3. However, Arl13 is equimolar to Arl3. Are similar results obtained when Arl13 is also used at the 1/20th concentration? This will provide even more context for understanding how Arl13 and BART function together as GEF and "co-GEF".

---

## [Author Response]

Essential revisions:As described in detail below, the authors should provide more quantitation: they should determine reaction rates and also provide statistics for replicates. Important will be to use a single nucleotide for the measurements as chemically distinct nucleotides will have different affinities (see below).

We have now provided quantification and determined the reaction rates. We also provided statistics for replicates. We have used single nucleotide measurements (GDP/GDP) to provide information about the rates (by using single nucleotide GDP/GDP we show that BART accelerates the exchange, in presence of ARL13b, 70 folds under the conditions used).

They should improve Figure 1 by providing more experiments, including GDP/GDP exchange and also the EDTA exchange reactions.

We have provided new experiments: Figure 1A addresses the extent of exchange at equilibrium where we show that at physiological GDP/GTP relative concentrations, majority of ARL3 will be majorly loaded with GDP in the presence of ARL13b. Figure 1B we have used equivalent nucleotides (GDP/GDP) and the experiments were recorded over long period of times until completion. EDTA does not accelerate the dissociation of GDP from ARL3 as has been published before (Hillig et al., 2000). Alternatively we have observed and recoded the exchange data over longer period of times, until completion, and provided this data.

They should improve Figure 3 by also testing reduced concentration of Arl13.

We used both BART and ARL13B at limiting amounts in the GEF reaction and clarified that in the text (Figure 4B).

They should improve the Discussion.

We have now rearranged the Discussion and elaborated on the points suggested by the reviewers and added some additional points.

Here are more Specific comments to guide your next steps:1) The authors often make (semi)quantitative statements based on their fluorescence curves and nucleotide release assays, yet they do not provide definite numbers that could be obtained from fitting the results (e.g. Figure 1A and B). This should be performed in order to make the effects of the experiments comparable to each other. In fact, I am not sure that there is an actual difference in nucleotide release rates in Figure 1A and B: The degree of exchange is certainly different for mantGDP and mantGppNHp and this can be explained by the different affinities since GDP binds ~50x more tightly to Arl3 than GppNHp. However, looking at the time it takes for the reactions to reach a plateau, they appear to do so at similar rates. A quantification and the use of equivalent nucleotides would be important to reliably estimate the effects.

We have now performed new experiments and replaced Figure 1A and B as follows: Figure 1A we show the equilibrium effect where we incubated ARL3GDP with different mixtures of nucleotides in the presence of ARL13B and reported the levels of exchange. This experiment shows that at equilibrium the extent of exchange is markedly affected by the difference in affinities between GDP and GTP and opens the question of how the cell overcomes this preference towards GDP. Figure 1B: We recorded the dissociation data using excess equivalent nucleotides (GDP/GDP) and non-equivalent nucleotides (to highlight the equilibrium preference towards GDP). We recorded the data over longer period of times, until completion, and report the rates to show the effect of ARL13B on accelerating the rates.

2)The Discussion should include a section describing the mechanism of action of other small GTPase GEFs with respect to the dependence on co-GEFs. In particular, the authors should discriminate between the definition of GEFs composed of several subunits and co-GEFs. What is the definition of a co-GEF? Is BART the first example of a co-GEF?

We have now provided this information in the Discussion and highlighted the difference between our proposed term of co-GEF and the known multimeric GEFs.

3) If they authors are trying to reproduce physiological GTP/GDP ratios, they should make sure that also unlabeled GDP is available in the buffer at its approximate physiological concentration.

We have now provided this in Figure 1A.

4) I am concerned about the use of non-equivalent nucleotides in the nucleotide-release assays. E.g., in Figure 1A the authors displace mant-GDP with GppNHp. Given that the affinities for GDP and GTP/GppNHp differ by a factor of ~50, equivalent nucleotides should be used for the displacement (e.g. displacing mant-GDP with GDP). Otherwise, the degree of exchange may be dependent on the relative affinities of GDP and GTP for the GTPase. Furthermore, full exchange/displacement should be indicated for every experiment (e.g. by adding EDTA and thus accelerating exchange).

We want to highlight the kinetic effect as well as the extent of exchange. We have clarified this now by providing a new figure, Figure 1A, which is focused on the degree of exchange at physiological GDP/GTP ratios and at equilibrium. We have also provided data for equivalent nucleotides (GDP for GDP) in Figures 1B and 3B). Finally we have now provided full reactions by recording the data over longer period of times when needed, as EDTA does not accelerate the exchange of bound GDP in case of ARL3, (Figures 1B, 3B and F). (Hillig et al., 2000).

5) Introduction: Describe lipidation of Arl. Explain what the physiological function of GDI is.

We have added this now in the text

6) Introduction: "In addition, the binder of ARL2 (BART) has been previously identified as an effector of ARL2 and ARL3": Please provide reference for this claim.

We have provided these references now (Sharer and Kahn, 1999, Veltel et al., 2008, Davidson et al., 2013).

7) Results: "The addition of recombinant ARL13B18-278…": Please explain why this construct has been used for the experiments. It is mentioned here for the first time. Provide reference to this construct if applicable.

The ARL13B^18-278^ construct was previously reported (Gotthardt et al., 2015) as a minimum active construct of ARL13B which could bind nucleotide-free ARL3. Using our *E. coli* expression system, longer constructs of ARL13B could not be successfully expressed and purified. We have added and cited this explanation.

8) Results: "GEFs are known not to favour the binding to, and hence dissociation of, one nucleotide-bound form of their cognate GTPase over another." Please provide reference for this claim.

We have provided this reference now (Goody, 2014).

9) Results: "Palmitoylated ARL13B is difficult to obtain due to its instability": Please explain in which sense Arl13B is unstable.

We meant the stability of the lipidation and the solubility of the protein. We have now added this clarification in the text

10) Results: "The addition of recombinant ARL13B18-278 to GDP-loaded ARL3 in the presence of up to 250-fold…" Please indicate properly when mant-nucleotides have been used for monitoring exchange reactions. In this instance, the use of mant-nucleotides is not indicated explicitly.

This figure was replaced and we made sure the use of nucleotides is indicated throughout.

11) Figure 2A: The quality of the depiction should be improved. In particular, the mix in ribbon (zig-zag loops) and cartoon representation is difficult to grasp. Also, light green and green depictions do not sufficiently discriminate the structures from each other.

The figure has been amended to focus only on the relevant regions undergoing conformational changes.

12) Figure 2C: Statistical errors and test of statistical significance should be included.

We have provided this now and modified the text accordingly.

13) Figure 3A: The input should be shown in order to evaluate potential changes of the proteins in the course of the experiment. It appears as if Arl3 is not pure, is this true? Furthermore, there is an additional band in the sample GDP+Alk. Phosphatase with GST-BART that disappears in the other lanes. Why is this? I also think that bar diagrams from such experiments should contain error bars. From the Coomassie-stained gel, there appears to be almost no difference in the intensities of Arl3 (if Arl3 is the band with lowest Mw), yet there are large differences in the WB. How can the authors explain this observation?

A panel showing input samples has been added to the figure. ARL3 is pure, the background bands arise from the GST-UNC119A preparation which cannot be further purified; this is more evident in the input lanes. The additional band in the AP reactions running just above ARL3 is GST; this is now clearly labelled. The quantification averages of three experiments are shown in the bar graph with their corresponding error bars. Comparing ARL3 band intensities on the Coomassie gel can be made difficult due to the presence of background bands, which are eliminated in the Western blots. We have re-run the gels with a higher sample volume, making the differences clearer.

14) Figure 3B: There appears to be a discrepancy with the figure legend since the figure mentions GppNHp but the text refers to GDP.

We have changed this figure now and changed the legend accordingly.

15) Figure 3C-H: Why have the experiments been conducted at differing BART and Arl13B concentrations? In my opinion, the concentrations should be kept comparable as much as possible (except for titrations, of course). Otherwise, the observed differences cannot be compared directly.

We have provided new experiments where we kept the concentrations comparable as much as possible (Figures 2 and 3).

16) Figure 5 appears to contain data. As such, it should not be part of the Discussion but part of the Results section.

We have amended this accordingly and included this figure in the Results section.

17) Discussion: "Due to the weak binding affinities and the fast dissociation of the bound nucleotide, it has been proposed that ARL2 does not need a GEF and undergoes rapid nucleotide exchange on its own.": Please provide reference for this claim.

We have clarified this sentence and provided reference now

18) There should be no "data not shown" in the manuscript.

In this experiment we have performed mass spectrometry analysis on the sample where we could not detect myristoylated peptides. We have now clarified in the text that we used mass spectrometry.

Reviewer #2:It would be nice to see some experimental evidence for the Arl3-Arl13B-BART complex modeled in Figure 3G. Alternatively, the authors could design point mutations to disrupt the ARL3-BART interaction and show an effect using their GEF assay.

We would like to thank the reviewer for this comment. We modelled our ternary complex using published individual binary crystal structures. Indeed, this model interpretation should be only restricted to the conclusion that a ternary complex cannot be structurally excluded. We hope in the future that we and others, can obtain structural data of the ternary complex.

Reviewer #3:1) A full understanding of the "physiological" nucleotide dependence of the exchange experiments shown in Figure 1 would include two additional comparisons:A) The experiment in Figure 1A should be repeated to show what happens if unlabeled GDP is added at time 0, as this addresses the question of whether GDP can efficiently replace mantGDP. Judging by the result shown in Figure 3E, GDP is not able to efficiently replace mantGDP. This result should be discussed in the first part of the Results section to more clearly explain that the kinetic observations of nucleotide preference are due to an inability of Arl13 to stimulate release GDP. Otherwise, the impression given from the results in Figure 1 is that GDP might be able to displace either mantGDP or mantGMP-PNP in the presence of Arl13.

We agree with the reviewer, excess GDP should be able to completely displace mantGDP. We have recorded the data over longer period of times and until completion and provided this data now in Figures 1B and 3B. The results address the reviewer’s comment as mentioned previously.

B) For some GTPases, GMP-PNP does not bind nearly as well as GTP. The unlabeled GMP-PNP experiments in Figure 1 should therefore be repeated using unlabeled GTP.

We have provided this experiment in Figure 1A.

2) In Figure 3, the GEF experiments are described as "multiple turnover" because BART is 1/20th the concentration of Arl3. However, Arl13 is equimolar to Arl3. Are similar results obtained when Arl13 is also used at the 1/20th concentration? This will provide even more context for understanding how Arl13 and BART function together as GEF and "co-GEF".

UNC119A-driven multiple turnover of BART is achieved with both BART and ARL13B at limiting amounts in the GEF reaction (Figure 4B).